# Periodontitis and pre-eclampsia among pregnant women in Rwanda: A case-control study

**Agnes Gatarayiha**[1] *, **Joseph Ntaganira**[2], **Zoe Brookes**[3], **Léon Mutesa**[4], **Anders Gustafsson**[5], **Stephen Rulisa**[6]

1 College of Medicine and Health Sciences, School of Dentistry, University of Rwanda, Kigali, Rwanda, 2 College of Medicine and Health Sciences, School of Public Health, University of Rwanda, Kigali, Rwanda, 3 Peninsula Dental School, University of Plymouth, Plymouth, United Kingdom, 4 Centre for Human Genetics, College of Medicine and Health Sciences, School of Medicine and Pharmacy, University of Rwanda, Kigali, Rwanda, 5 Karolinska Institutet, Stockholm, Sweden, 6 College of Medicine and Health Sciences, School of Medicine and Pharmacy, University of Rwanda, Kigali, Rwanda

* agnesgata31@gmail.com

**Data Availability Statement:** Data Availability Statement The data that support the findings from this study are publicly available at Zenodo via https://zenodo.org/records/10928051.

## Abstract

### Introduction

Several studies have indicated that the presence of periodontitis during pregnancy could increase the risk of developing pre-eclampsia, thereby negatively influencing pregnancy outcomes for both the mother and child. Notably, despite the high prevalence of both periodontitis and adverse pregnancy outcomes in Rwanda, there exists a crucial evidence gap concerning the precise relationship between periodontitis and pre-eclampsia.

### Objectives

The aim of this study was to assess the association between periodontitis and pre-eclampsia amongst pregnant women in Rwanda.

### Methods and materials

Employing an unmatched 1:2 case-control design, we studied 52 pre-eclamptic and 104 non-pre-eclamptic pregnant women aged $\geq$18 years at two referral hospitals in Rwanda. Pre-eclampsia was defined as a systolic blood pressure $\geq$ 140 and diastolic blood pressure $\geq$ 90 mm Hg, diagnosed after 20 weeks of gestation and proteinuria of $\geq$300mL in 24 hours of urine collection. Periodontitis was defined as the presence of two or more teeth with one or more sites with a pocket depth $\geq$ 4mm and clinical attachment loss >3 mm at the same site, assessed through clinical attachment loss measurement. Bivariate analysis and logistic regression were used to estimate Odds ratio (ORs) and 95% confidence interval.

### Results

The prevalence of periodontitis was significantly higher among women with pre-eclampsia, compared to pregnant women without pre-eclampsia, at 90.4% and 55.8%, respectively (p<

**Funding:** This research received funding from the Capacity Building for Female Scientists in East Africa Africa program (CaFe-SEA) under the East African Consortium for Clinical Research (EACCR) partner's institution funded by EDCTP and University of Rwanda-Sweden collaboration. These had no other involvement for this study.

**Competing interests:** The authors have declared that no competing interests exist.

0.001). Pregnant Women with periodontitis were 3.85 times more likely to develop pre-eclampsia after controlling for relevant confounders (adjusted Odds Ratio [aOR] = 3.85, 95%CI = 1.14–12.97, p<0.05).

## Conclusion

This study results indicates that periodontitis is significantly associated with pre-eclampsia among pregnant women in Rwanda. These findings suggest that future research should explore whether enhancing periodontal health during pregnancy could contribute to reducing pre-eclampsia in this specific population.

## Introduction

Pre-eclampsia is a major complication of pregnancy characterized by gestational hypertension and associated with significant maternal and fetal risks, including impaired fetal development, maternal morbidity and mortality [1]. Globally, the prevalence of pre-eclampsia ranges from 2% to 8% of pregnancies [2, 3], with a prevalence of 4.1% in developing countries like Sub-Saharan Africa [4]. In Rwanda, recent studies have reported a prevalence of 2.3% in Kigali teaching hospitals [5] and approximately 3% in rural areas [6], highlighting the ongoing challenge posed by this condition.

According to the American College of Obstetrics and Gynecology (ACOG), pre-eclampsia is defined as hypertension ($\geq$ 140/90 mm Hg) and proteinuria (300 mg or more per 24-hour urine collection) or, in the absence of proteinuria, new-onset hypertension with associated complications, such as thrombocytopenia, renal insufficiency, impaired liver function, pulmonary edema, severe headache unresponsive to medication, or visual disturbances [3, 7]. This condition typically manifests after 20 weeks of gestation [3, 7], underscoring the importance of early detection and management.

Pre-eclampsia contributes to approximately 50,000 maternal deaths globally each year [8]. In low- and middle-income countries, it is responsible for 10–15% of maternal deaths [9], with Rwanda experiencing a rate of 13% [10]. The condition's multifactorial etiology includes factors such as endothelial dysfunction, systemic inflammation, and various risk factors like primiparity, family history, uterine malformation, pre-existing hypertension, diabetes, obesity, placental abnormalities, renal disease [11, 12]. Recent research has highlighted the role of systemic inflammation and endothelial dysfunction in the pathogenesis of pre-eclampsia [13–15]. Systemic inflammation, often exacerbated conditions such as periodontitis, has been proposed as a key mechanism contributing to the development of pre-eclampsia [14, 16].

Periodontitis, a chronic inflammatory disease caused by localized bacterial infections, results in the destruction of periodontal tissues and can lead to tooth loss [17, 18]. Globally, severe periodontitis affects approximately 616 million people [19, 20], with a higher prevalence of 13.5% reported in central Sub-Saharan Africa [21]. In Rwanda, a national oral health survey found a high prevalence of periodontitis in general population, with 60% of people experiencing dental calculus and 34.2% having dental debris [22]. Furthermore, periodontitis is found in up to 40% of pregnant women [23], and a growing body of evidence suggests a compelling link between periodontitis and adverse pregnancy outcomes, including pre-eclampsia [24–27]. Studies have reported associations with adverse pregnancy outcomes such as preterm births, low birth weight and pre-eclampsia [28–30]. For instance, research in Korea found a significant association between periodontitis and an increased risk of pre-eclampsia [31], while

studies in Tanzania revealed a substantial association with high odds of pre-eclampsia (aOR = 4.12; 95% CI: 2.20–7.90) [32].

The mechanistic relationship between periodontitis and pre-eclampsia involves several key pathways. Chronic systemic inflammation resulting from periodontitis may exacerbate the inflammatory processes involved in pre-eclampsia. Periodontal disease can lead to elevated levels of inflammatory markers such as C-reactive protein (CRP), interleukin-6 (IL-6), and tumor necrosis factor-alpha (TNF-α), which may contribute to endothelial dysfunction and impaired placental blood flow, thereby increasing the risk of pre-eclampsia [33, 34]. This connection underscores the need for further research to explore these inflammatory markers and their role in the pathogenesis of pre-eclampsia. Systematic review from 30 studies confirms that pre-eclampsia is a significant risk factor for periodontitis, and even suggests that is more pronounced in lower-middle-income countries [35, 36].

Despite the growing body of evidence, most research on the relationship between periodontal disease and pre-eclampsia has been in developed countries, creating a significant knowledge gap for low and middle income countries. In Rwanda, where both pre-eclampsia and periodontal disease are prevalent, there is a need for localized studies to better understand this association. This study aims to investigate the association between periodontitis and pre-eclampsia in Rwanda. By doing so, it seeks to provide valuable insights into how these conditions interact in the context of Rwanda, contributing to improved health outcomes and advancing of knowledge in this critical area.

## Methods and materials

### Study design

An unmatched 1:2 case control design was used to comprehensively investigate the association between periodontitis and pre-eclampsia among pregnant women in Rwanda. The case group comprised pregnant women diagnosed with pre-eclampsia, while the control group consisted of normotensive pregnant women.

### Sample and procedure

To determine the sample size, we utilized G* Power Software, aiming for a significance level (α \alpha) of 0.05 and a target power of 85%, which is a common standard for many studies. to determine the necessary number of participants. The effect size was set at 0.50, representing a medium to large odds ratio (OR) of approximately 2.61. Based on this effect size and power level, the required total sample size was 156 participants, with 52 women with pre-eclampsia (case) and 104 women without pre-eclampsia (control) [37], (Fig 1). Pre-eclampsia, in this context, was defined as a systolic blood pressure $\geq$ 140 mm Hg and diastolic blood pressure $\geq$ 90 mm Hg diagnosed after 20 weeks of gestation, in women who were normotensive previously. Pre-eclamptic women also had proteinuria, with of urine protein concentration of $\geq$ 300 mg over 24 hours urine collection [7, 38].

For the inclusion criteria, this study includes singleton pregnant women at 20 weeks' gestation or beyond, aged 18 years and older, and with non-history of hypertension previously before the current pregnancy. Furthermore, selection was limited to women attending antenatal care services or admitted to maternity wards at designated hospitals (i.e., The University teaching Hospital of Kigali (CHUK) and Ruhengeri Hospital) during the study period (August 1, 2021, to February 25, 2022). On the other hand, the exclusion criteria included women with a history of placental abnormalities, or recent periodontal treatment requiring antibiotic or antiseptic mouthwashes during the 4 weeks preceding the study. However, women with their first pregnancy were not excluded in our study despite if they were diagnosed by healthcare

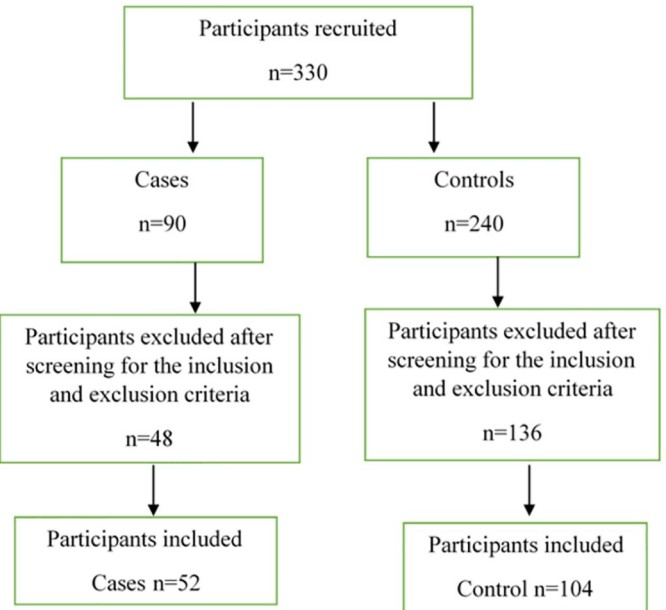

**Fig 1. A flow diagram of recruitment process for included participants.**

providers with any potential placental abnormalities during ante natal care after a thorough initial assessment during medical history, physical examination, and laboratory test results. Therefore, they were included unless present with such identification placental abnormalities during assessment. Furthermore, for periodontal treatment requiring antibiotic or antiseptic mouthwashes, 4-weeks period is usually used as a standard (practical approach, and is reported in other studies [39], because it allows adequate time for any residual effect of antibiotics or mouthwashes to dissolve. Antibiotics and mouthwashes can both can alter oral microbiome/bacteria, affecting periodontal health, hence inclusion could potentially affect the study outcomes.

Participants with dentures or orthodontic braces, and those with a history of in vitro fertilization were also excluded. Orthodontic braces themselves are not directly associated with preeclampsia, however, potential bad oral hygiene related to them is crucial, as this may cause chronic inflammation from the accumulated oral bacteria, which in turn could contribute to the development of preeclampsia during pregnancy. Thus we excluded patients with orthodontic braces during our study. This stringent selection process aims to ensure a homogeneous study population, minimizing potential confounding factors.

Our study adhered strictly to ethical guidelines, ensuring participant well-being and rights. Ethical clearance was granted by the Rwanda National Ethics Committee (Approval No. 154/RNEC/2021). Participants provided informed consent, fully briefed on the study's aims, potential risks, and benefits. They retained the right to withdraw at any time without affecting their medical care, underscoring our commitment to their autonomy and the confidentiality of their responses.

## Data collection measurements and procedure

The data collection was conducted by two calibrated dental therapists who was blinded to the study group allocations of the participants. They conducted personalized interviews to gather

comprehensive information on the participants' medical histories and socio-demographic characteristics, smoking and alcohol usage, and specific pre-eclampsia details including blood pressure, antihypertensive drug dosage, protein concentration and maternal clinical features such as oedema, headache, and visual disturbance. Simultaneously, physical examinations, encompassing blood pressure measurements and details on proteinuria were conducted during obstetrics and gynecology consultation. For the detection of proteinuria, Urine Protein Test Strips by Siemens Healthcare Diagnostics (Tarrytown, NY, USA) were employed. These were also obtained through local medical suppliers in Rwanda to measure the presence of protein in urine samples. This information was sourced from participants' medical records by trained clinician/nurse at the study hospitals.

## Periodontal measurement

For the periodontal examination, we assessed all teeth (excluding third molars) for the case and control groups. Periodontal measurements included the recording of probing pocket depth (PPD), gingival recession and clinical attachment loss (CAL) in millimeters (mm) at six sites surrounding each tooth (buccal, mesio-buccal, distal-buccal, lingual, mesio-lingual, and disto-lingual), using the University of North Carolina (UNC)-15 probe manufactured by Hu-Friedy (Chicago, IL, USA). The PPD was measured as the distance from the gingival margin to the apical portion of the gingival sulcus. CAL was derived from the aggregate of gingival recession and PPD measures, though gingival recession itself was not independently measured. Additionally, the measurement from the cemento-enamel junction (CEJ) to the base of the pocket/sulcus was taken to ascertain clinical attachment loss.

## Classification criteria for periodontitis severity

In defining periodontitis for this study, involvement had to include two or more teeth, each with at least one site demonstrating a probing pocket depth $\geq$ 4mm and clinical attachment loss >3 mm at the same site [14]. Reflecting on the constraints imposed by the absence of radiographs and building upon the methodology of previous similar studies [40, 41], we classified mild periodontitis as CAL of 4–5 mm and moderate/severe CAL $\geq$ 6 mm across at least two different teeth. This classification scheme enabled the safe and precise categorization of periodontitis severity among our participants without resorting to the use of radiograph. Due to the concerns for the safety of pregnant women participating in our study, we did not perform radiographic examinations for the purpose of research. This decision was made to protect the participants as recommended by the ethic committee and to maintain the ethical integrity of our research findings. Our approach acknowledges the challenges of applying an internationally recognized specific diagnosis of periodontal disease without such imaging, as might be recommended in guidelines from entities like the European Federation of Periodontology (EFP) consensus [42].

## Data analysis

All Analyses were conducted using SPSS (Statistical Package for the Social Sciences) Version 25. Binary variables including location, smoking, and alcohol consumption status, and medical pregnancy history were described using the frequencies and percentages. In contrast, continuous variables such as age, and weight, were described using mean and standard deviation.

A chi-square test was conducted for bivariate analysis to assess whether the difference in terms of proportion between dependent variables (pre-eclampsia) and other independent variables (periodontitis, socio demographic characteristics, medical, dental, and pregnant

conditions) are statistically significant. Additionally, the Mann-Whitney U test was employed to assess the difference in mean ranks for continuous variables between pre-eclamptic and non-preeclamptic women due to the non-normal distribution of the data.

Logistic regression analysis was conducted to examine the association between potential risk factors and pre-eclampsia while controlling for potential confounders. The covariates were chosen based on a purposeful selection process begun by a univariate analysis of each variable. Covariates were included based on results from Wald tests; those with p-values below.25 were retained. The results were reported as crude odd ratios (cOR) and adjusted odd ratios (aORs) with their 95% confidence intervals. All statistically significant variables in a bivariate analysis were included in the logistic regression model. P-values of < 0.05 were considered as statistically significant.

## Results

### Socio-demographic characteristics

This study included 156 pregnant women, comprising 52 women with pre-eclampsia and 104 women without the condition. The mean age of all participants was 30.4 years (SD = 4.9), with significant age differences observed between non-pre-eclamptic (29.6 years) and pre-eclamptic (32.0 years) groups (P = 0.01). The gestational age averaged 30.7±19.6 weeks across the cohort.

Regarding pregnancy and health history, 17.9% of participants were in their first pregnancy, 75% had 2–4 pregnancies and 7.1% had more than four pregnancies (Table 1). The average weight of women in pre-eclamptic group was significantly higher than in the non-pre-eclamptic group (74.5±9.2 kg versus 68.5±6.3 kg). A majority (79.5%) attended fewer than four antenatal care visits. Additionally, 8.7% of the non-pre-eclamptic group and 25.0% of the pre-eclamptic group reported alcohol consumption (P = 0.015).

Regarding smoking status, 2.8% of non-pre-eclamptic women were smokers compared to none in the pre-eclamptic group (P = 0.358). Although smoking rates were low, this difference may warrant further investigation. Information on oral hygiene revealed that 3.8% of non-pre-eclamptic women and 13.5% of pre-eclamptic women had visited a dental service in the previous 6 months (P = 0.027). Only a small fraction of women in both groups visited for scaling teeth (2.9% vs. 9.6%, P = 0.072).

### Comparison of periodontal parameters between pre-eclamptic and non-pre-eclamptic group

The analysis, conducted using the Mann-Whitney test for non-normally distributed continuous variables (PPD and CAL) and the chi-square test for categorical variables, revealed significant difference in periodontal health between groups with and without pre-eclampsia (Table 2). Compared to the non-pre-eclamptic group, women with pre-eclampsia significantly higher mean PPD (4.2±1.1 mm vs. 3.62±0.8 mm) and CAL (5.2±1.4 mm vs. 4.0±1.1 mm), with p<0.0001 for both measures. Additionally, the incidence of periodontitis was markedly greater in the pre-eclamptic population (90.4%) compared to the non-pre-eclamptic group (55.8%, p<0.001). Specifically, moderate to severe periodontitis (CAL >6 mm) was significantly more common in the pre-eclamptic group (42.3%) than in the non-pre-eclamptic group (13.5%, p<0.001). Furthermore, bleeding on probing was more prevalent in pre-eclamptic women (88.5%) as compared to non-pre-eclamptic women (65.4%), with this finding nearing statistical significance (p<0.056).

**Table 1. Demographic characteristics, pregnancy related, dental and medical data of participants.**

| Variables | Non-pre-eclamptic | Pre-eclamptic | P value |
|---|---|---|---|
| Age [**mean (SD)**] | 29.6(5.0) | 32.0(4.5) | 0.011 |
| Gestation in weeks [**mean (SD)**] | 30.6(4.6) | 31.2(3.8) | 0.464 |
| Weight for pregnant women in Kg [**mean (SD)**] | 68.5(6.3) | 74.5(9.2) | <0.001 |
| Antenatal care visits [**mean (SD)**] | 2.8(0.9) | 3.0(0.7) | 0.175 |
| **Age** | | | 0.004 |
| 18–28 | 47(45.2) | 10(19.2) | |
| 29–39 | 54(51.9) | 38(73.1) | |
| 40–59 | 3(2.9) | 4(7.7) | |
| **Weight** | | | 0.855 |
| 50–59 | 8(7.7) | 2(3.8) | |
| 60+ | 96(92.3) | 50(96.2) | |
| **Location [n (%)]** | | | 0.001 |
| CHUK | 39(35.5) | 34(65.4) | |
| RUHENGERI Hospital | 65(62.5) | 18(34.6) | |
| **Alcohol consumption [n (%)]** | | | 0.015 |
| Yes | 9(8.7) | 13(25.0) | |
| Stopped | 2(1.9) | 2(3.8) | |
| Non | 93(89.4) | 37(71.2) | |
| **Smoking status [n (%)]** | | | 0.358 |
| smoker | 3(2.8) | 0(0.0) | |
| stopped | 100(96.2) | 52(100.0) | |
| Non | 1(1.0) | 0(0.0) | |
| **Diabetes history [n (%)]** | | | 0.013 |
| Yes | 0(0.0) | 3(5.8) | |
| No | 104(100.0) | 49(94.2) | |
| **Other cardiovascular diseases [n (%)]** | | | 0.013 |
| Yes | 0(0.0) | 3(5.8) | |
| No | 104(100.0) | 49(94.2) | |
| **Parity [n (%)]** | | | |
| Less than one | 24(23.1) | 4(7.7) | 0.022 |
| Between 2 and 4 | 71(68.3) | 46(88.5) | |
| Above 4 | 9(8.7) | 2(3.8) | |
| **ANC [n (%)]** | | | 0.575 |
| less than 4 | 84(80.8) | 40(76.9) | |
| 4 and above | 20(19.2) | 12(23.1). | |
| **Past history of placenta abnormalities [n (%)]** | | | 0.377 |
| Yes | 3(2.9) | 3(5.8) | |
| No | 101(97.1) | 49(94.2) | |
| **Edema [n (%)]** | | | <0.001 |
| No | 90(86.5) | 23(44.2) | |
| Yes | 14(13.5) | 29(55.8) | |
| **Headache [n (%)]** | | | <0.001 |
| No | 94(90.4) | 22(42.3) | |
| Yes | 10(9.6) | 30(57.7) | |
| **Visited dental service in previous 6 months [n (%)]** | | | 0.027 |
| Yes | 4(3.8) | 7(13.5) | |
| No | 100(96.2) | 45(86.5) | |

(*Continued*)

**Table 1.** (Continued)

| Variables | Non-pre-eclamptic | Pre-eclamptic | P value |
|---|---|---|---|
| **Visited the dental service for scaling teeth [n (%)]** | | | 0.072 |
| Yes | 3(2.9) | 5(9.6) | |
| No | 101(97.1) | 47(90.4) | |

**Abbreviations:** SD, standard deviation; n, frequency; %, percentage; CHUK, Centre Universitaire Hospitalier de Kigali; ANC, Antenatal care visits; Kg, Kilograms.

**Table 2. Periodontal conditions between pre-eclamptic and non-pre-eclamptic groups.**

| Variables | Non-pre-eclampsia | Pre-eclampsia | P-value |
|---|---|---|---|
| | (N = 104) | (N = 52) | |
| **Periodontal conditions** | | | |
| Bleeding on probing (%) | 68 (65.4) | 46 (88.5) | 0.056 |
| Periodontitis, (%)* | 58 (55.8) | 47 (90.4) | <0.001 |
| Mild: 4 to 5 mm CAL (%)* | 44 (42.3) | 25 (48.1) | 0.494 |
| Moderate/severe >6 mm CAL (%)* | 14 (13.5) | 22 (42.3) | <0.001 |
| **Clinical parameters (mean—SD)** | | | |
| Periodontal pocket depth, mm ** | 3.62±0.8 | 4.2±1.1 | <0.0001 |
| Clinical attachment loss, mm ** | 4.0±1.1 | 5.2±1.4 | <0.0001 |

Abbreviations:

*significant chi-square ($X^2$) test;

**significant Mann Whitney test;

CAL, clinical attachment loss, mm, millimeters.

## Factors associated with pre-eclampsia

The chi-square significantly showed differences between the groups in the variables including age, weight, study location, alcohol consumption, smoking, diabetes, periodontal Health Variables, cardiovascular disease, parity, edema, headache, visual disturbances, use of anti-hypertensive medication. Table 3 presents the findings from a binary logistic regression analysis exploring the association between periodontitis and pre-eclampsia among pregnant women. Initial crude analysis showed that women with periodontitis were 7.5 times more likely to develop pre-eclampsia (OR = 7.46, 95%CI = 2.74–20.26; p<0.05). This association remained significant after adjusting for confounders (aOR = 3.85, 95% CI = 1.14–12.97, p<0.05), indicating that pregnant women with periodontitis were approximately 3.85 times more likely to develop pre-eclampsia than those without periodontitis, after adjusting for confounders. For moderate/severe periodontitis, the association with pre-eclampsia was significant in the unadjusted model (cOR: 4.71, 95% CI: 2.15–10.36, p<0.05) but became non-significant after adjustment for confounders (aOR: 1.74, 95% CI: 0.68–5.27). Crude analysis also demonstrated that other risk factors such as study location, alcohol consumption, age, dental services visits in the previous 6 months, primiparity, experiencing edema and headache were associated with pre-eclampsia. Notably, variables such as study site location, experiencing edema, and headache also remained significantly associated with pre-eclampsia after controlling for confounders.

**Table 3. Binary logistical regression analysis for association between periodontitis and other risk factors, and pre-eclampsia among pregnant women.**

| Study variables | Model 1 | Model 2 |
|---|---|---|
| | cOR | aOR |
| **1. Periodontitis condition** | | |
| **Periodontitis** | | |
| No | 1 | 1 |
| Yes | 7.46(2.74–20.26) * | 3.85(1.14–12.97) * |
| **Moderate/severe periodontitis** | | |
| No | 1 | 1 |
| Yes | 4.71(2.15–10.36) * | 1.74(0.68–5.27) |
| **2. Other associated risk** | | |
| **Site location** | | |
| CHUK | 3.15(1.57–6.31) * | 2.35(0.87–6.35) |
| Ruhengeri district Hospital | 1 | 1 |
| **Alcohol consumption** | | |
| No | 1 | 1 |
| Yes | 0.28(0.109–0.70) * | 0.35(0.09–1.25) |
| **Visited dental service in previous 6 months** | | |
| No | 1 | 1 |
| Yes | 3.88(1.08–13.96) * | 4.67(0.91–24.12) |
| **Oedema** | | |
| No | 1 | 1 |
| Yes | 8.11(3.70–17.78) * | 3.61(1.17–11.105) |
| **Headache** | | |
| No | 1 | 1 |
| Yes | 12.82(5.46–30.08) * | 3.12(0.97–10.02) |
| **Parity** | | |
| 1 or less | 1 | 1 |
| 2–4 | 3.27(1.26–11.93) * | 1.71(0.35–8.371) |
| Above 4 | 1.33(0.21–8.58) | 0.98(0.08–12.04) |
| **Age** | | |
| 18–28 years | 1 | 1 |
| 29–39 years | 3.31(1.49–7.35)* | 1.45(0.46–4.59) |
| 40–59 years | 6.27(1.21–32.48)* | 6.38(0.68–59.87) |

**Abbreviations:** 1, reference category;

*statistically significant (p<0.05);

cOR, crude odds ratio; aOR, adjusted odds ratio.

## Discussion

The findings of this study support the hypothesis that periodontitis is associated with pre-eclampsia in two selected referral hospitals in Rwanda. Our observation of a notably higher prevalence of periodontitis (90.4%), including moderate/ severe periodontitis cases, among women with pre-eclampsia aligns with several global studies. For instance, a study done in India found a similar association, where women with periodontitis had a significantly higher risk of developing pre-eclampsia standing at 90% [43]. This pattern is also observed in studies from Iran at 98% [15] and Turkey at 74.3% [44], reinforcing the potential global relevance of this association.

The prevalence of periodontitis in our study population significantly exceeded that of the general adult population in Rwanda (39.6%), [45].This elevated prevalence among pre-eclamptic women mirrors findings from studies in the USA, where pregnant women with periodontitis were found to have higher rates of pre-eclampsia compared to those without periodontitis [16]. Similarly, a Tanzanian study demonstrated a strong association between periodontal disease and adverse pregnancy outcomes, including pre-eclampsia [32].

In contrast, some studies have reported conflicting results. For example, a study conducted in Norway found no significant association between periodontitis and pre-eclampsia after adjusting for confounders [46]. Additionally, a study from Japan reported similar findings, where the link between periodontal disease and pre-eclampsia was not statistically significant [47]. These discrepancies could be due to differences in study populations, periodontal disease definitions, or variations in healthcare access and practices across different countries.

Our multivariate logistical regression analysis was adjusted for potential confounders such as age, alcohol consumption, primiparity, edema, and headache. The resulting odds ratio of 3.85, indicates that women with moderate/severe periodontitis are over three times more likely to develop pre-eclampsia, which is consistent with a meta-analysis that reported an overall odds ratio of 3.18 [35]. Interestingly, this meta-analysis found that the association was even stronger in studies conducted in low- and middle-income countries (OR = 6.70), [35], underscoring the importance of context-specific research.

In terms of clinical indicators, our study found significantly higher mean periodontal pocket depth (PPD) and clinical attachment loss (CAL) in the pre-eclamptic group, which aligns with the findings from studies in Brazil and India. In Brazil, researchers observed that pre-eclamptic women had greater PPD and CAL compared to their non-pre-eclamptic counterparts [48]. Similarly, an Indian study highlighted significant differences in periodontal measurements between pre-eclamptic and non-pre-eclamptic women, further supporting the association between periodontal health and pregnancy outcomes [49]. These findings suggest that the severity of periodontal disease may be a critical factor in the development of pre-eclampsia.

Univariate analysis in our study identified association between pre-eclampsia and various risk factors, including age, study location, alcohol consumption during pregnancy, visit to dental services, edema, headache, primiparity, pregnant women's weight and visual disturbances.

The final multivariate logistic regression confirmed age, study location, edema and headache as significant risks factors for pre-eclampsia. These results are consistent with findings from studies done in Chile and Bangladesh, which identified similar risk factors for pre-eclampsia, including maternal age and pre-existing hypertension [50, 51]. Additionally, studies in Sweden and South Africa emphasized the role of socio-economic factors in pre-eclampsia risk [52, 53], suggesting that future research in Rwanda should also consider these variables.

Interestingly, our study found a significant association between dental service visits and pre-eclampsia, which contrasts with findings from Boggess et al, that reported no significant link between dental visits and pre-eclampsia [54]. This difference might be explained by varying access to dental care and differences in dental service utilization during pregnancy across different healthcare systems. Furthermore, our study did not find an association between tobacco consumption and pre-eclampsia, a finding that diverges from studies in Sweden and the USA, which reported smoking as a significant risk factor for pre-eclampsia [48, 55].

Potential mechanisms for the link between periodontitis and pre-eclampsia include the translocation of periodontal pathogens to the placenta, leading to localized inflammation and oxidative stress, which may contribute to endothelial dysfunction—a key feature of pre-eclampsia [35]. This theory is supported by studies that detected periodontal bacteria in

placental tissues of pre-eclamptic women, suggesting a direct pathogenic role [35, 56]. More-over, the hypothesis that repeated bacteremia exposes decidual tissues to periodontal bacteria, has been proposed as another pathway, although the exact mechanism remain to be fully eluci-dated [15, 50]. Despite these plausible explanations, the multifactorial nature of both condi-tions necessitates further research to clarify these pathways and determine whether improving periodontal health can reduce pre-eclampsia risk.

## Strength and limitations of our study

Our study is the first to explore association between periodontitis and pre-eclampsia among pregnant women in Rwanda, offering valuable insights into this important public health issue. However, the small sample size and omission of certain socio-demographic characteristics may limit the generalizability of our findings. Future research should aim to collect individual-level data on socioeconomic factors to deepen our understanding. Additionally, the absence of dental radiographs due to the safety concern, a limitation noted in similar studies [40–42], restricted our ability to assess of bone loss radiographically and provide a universally accepted diagnosis of periodontal disease. While we adjusted for known confounders in our multivari-ate logistic regression model, residual confounding may still exist due to the lack of detailed endothelial function measures or accurate quantifications of placental insufficiency. These lim-itations highlight the need for future studies to include larger sample sizes, comprehensive socio-economic data, additional inflammatory markers, and advanced diagnostic methods to strengthen the observed associations and provide a more robust understanding of the relation-ship between periodontitis and pre-eclampsia.

## Conclusion

This study establishes a robust association between periodontitis and pre-eclampsia among pregnant women in Rwanda, even after accounting for potential confounders. Our findings indicate that pregnant women with periodontitis are approximately three times more likely to develop pre-eclampsia compared to their counterparts. This significant association under-scores the impact of periodontal health on maternal outcomes and highlights the importance of integrating oral health care into prenatal care practices, as interventions to manage peri-odontitis once pregnancy may be too late; non-surgical therapy for example does not reduce adverse pregnancy outcomes [57].

Despite some limitations, our research contributes valuable insights into the association between periodontal disease and pre-eclampsia. The evidence suggests that improving oral health could be a viable strategy to reduce the risk of pre-eclampsia, emphasizing the need for preventive interventions. The findings have important implications for maternal and fetal health, particularly in resource-limited settings where access to comprehensive dental care may be limited. Future research should address regional variations in healthcare access, socio-economic factors, and the definitions of periodontal disease to better understand these relationships.

## Supporting information

**S1 Checklist. STROBE statement—Checklist of items that should be included in reports of *case-control studies.***
(DOC)

## Acknowledgments

The authors sincerely wish to thank all women for their kind cooperation. Authors would wish to recognize the research assistants for their contributions during data collection.

## Author Contributions

**Conceptualization:** Agnes Gatarayiha, Joseph Ntaganira, Zoe Brookes, Léon Mutesa, Anders Gustafsson, Stephen Rulisa.

**Data curation:** Agnes Gatarayiha.

**Formal analysis:** Agnes Gatarayiha, Zoe Brookes, Léon Mutesa, Anders Gustafsson, Stephen Rulisa.

**Funding acquisition:** Agnes Gatarayiha.

**Investigation:** Agnes Gatarayiha.

**Methodology:** Agnes Gatarayiha, Joseph Ntaganira, Zoe Brookes, Léon Mutesa, Anders Gustafsson, Stephen Rulisa.

**Project administration:** Agnes Gatarayiha.

**Supervision:** Joseph Ntaganira, Zoe Brookes, Léon Mutesa, Anders Gustafsson, Stephen Rulisa.

**Writing – original draft:** Agnes Gatarayiha.

**Writing – review & editing:** Agnes Gatarayiha, Joseph Ntaganira, Zoe Brookes, Léon Mutesa, Anders Gustafsson, Stephen Rulisa.

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
