## [Decision Letter · Decision Letter 0]

26 Feb 2024

PONE-D-23-41485Periodontitis and Pre-eclampsia among Pregnant Women in Rwanda: A Case-Control Study.PLOS ONE

Dear Dr. Gatarayiha,

Thank you for submitting your manuscript to PLOS ONE. After careful consideration, we feel that it has merit but does not fully meet PLOS ONE’s publication criteria as it currently stands. Therefore, we invite you to submit a revised version of the manuscript that addresses the points raised during the review process.

We look forward to receiving your revised manuscript.

Kind regards,

Offer Erez, M.D.

Academic Editor

PLOS ONE

Journal Requirements:

3. Please expand the acronym “EDCTP” (as indicated in your financial disclosure) so that it states the name of your funders in full.

"The authors sincerely wish to thank all women for their kind cooperation. Authors would wish to recognize the research assistants for their contributions during data collection. We also acknowledge support from the Capacity Building for Female Scientists in East Africa program (CaFe-SEA) under the East African Consortium for Clinical Research (EACCR) partner’s institution funded by EDCTP and University of Rwanda-Sweden Collaboration."

"This research received funding from the Capacity Building for Female Scientists in East Africa Africa program (CaFe-SEA) under the East African Consortium for Clinical Research (EACCR) partner’s institution funded by EDCTP and University of Rwanda-Sweden collaboration. These had no other involvement for this study. "

5. We note that you have indicated that there are restrictions to data sharing for this study. For studies involving human research participant data or other sensitive data, we encourage authors to share de-identified or anonymized data. However, when data cannot be publicly shared for ethical reasons, we allow authors to make their data sets available upon request. For information on unacceptable data access restrictions, please see http://journals.plos.org/plosone/s/data-availability#loc-unacceptable-data-access-restrictions. 

6. Please include a caption for figure 1. 

7. Please ensure that you refer to Figure 1 in your text as, if accepted, production will need this reference to link the reader to the figure.

8. Please include your tables as part of your main manuscript and remove the individual files. Please note that supplementary tables (should remain/ be uploaded) as separate ""supporting information"" files

Reviewers' comments:

Reviewer's Responses to Questions

**Comments to the Author**

1. Is the manuscript technically sound, and do the data support the conclusions?

Reviewer #1: Yes

Reviewer #2: Yes

2. Has the statistical analysis been performed appropriately and rigorously? 

Reviewer #1: Yes

Reviewer #2: Yes

3. Have the authors made all data underlying the findings in their manuscript fully available?

Reviewer #1: No

Reviewer #2: Yes

4. Is the manuscript presented in an intelligible fashion and written in standard English?

Reviewer #1: Yes

Reviewer #2: No

5. Review Comments to the Author

Reviewer #1: Manuscript Review Comments

Title: “Periodontitis and Pre-eclampsia among Pregnant Women in Rwanda: A Case-Control

Study.” (ID: PONE-D-23-41485).

The authors present a case-control study in which they address the association between periodontitis and preeclampsia. Although it is an interesting topic, this reviewer has doubts about several aspects presented in the manuscript.

Comment to the authors

- Women with preeclampsia would have 3.85 times more chances of having periodontitis compared to women without preeclampsia, not the opposite, considering this study design (as stated in the abstract). Please consider this throughout the whole manuscript.

- Revise that keyword are Mesh terms for better indexation (alcohol consumption).

- Old numbers of periodontitis prevalence? 2010 (Introduction)

- There are meta-analysis/systematic reviews on perio-preeclampsia that have not been cited in this manuscript. Please check the work by Quynh-Anh Le et al. (2022).

- Indicate the commercial reference of all materials and product used in the study (including software).

- The definition of the study groups is the diagnosis (or not) of preeclampsia, the presence of periodontitis was assessed later and did not have relevance in these groups, so I would recommend removing it from the sentence to avoid confusion to the reader. Also, this group definitions, as well as the definition of preeclampsia should go in study design.

- I would suggest the author to follow the STROBE guidelines to provide the best structure to report these kinds of studies.

- The correct term is probing pocket depth (PPD), not PD. Please replace it.

- The definition of periodontitis used by the authors refers to an article from 2012. Please refer to the most recent case definition and classification of periodontitis from the EFP workshop of 2018.

- Why was Mann-Whitney U test used for continuous variables? Did the authors assess normality of the data?

- Tables are presented as supporting information. Is this correctly done?

- Tables should provide units of measure of each variable and the statistical tests used in each case, indicated in table footnotes.

- What do the column “study sample” contain? The total of participants from both groups? If that, what does this column provide as an information for the reader? It may be confusing.

- Table 2: “<0.056” is not a relevant p-value and the sign < may be an error.

- I think periodontitis severity should be better presented following the staging and grading classification, rather than the one provided by the authors.

- Table 3: Please indicate the adjustment variables included in the model 2 and justify why they were selected as potential confounders.

- Conclusions stated by the authors are too extensive, please rephrase them in a more concise way, giving answers to the aims of the study.

Reviewer #2: Dear, Authors

It’s my great pleasure to be a reviewer of this research that focuses on “Periodontitis and Pre-eclampsia among Pregnant Women in Rwanda: A Case-Control Study,” which has great public health importance. But, I do have some comments and suggestions on this research paper as detailed below.

1. Affiliation: better to include “city” where the university is located, for example, school of Dentistry, xxxxx, Rwanda. Plus, 3rd and 5th make full details if possible

2. Abstract [introduction part]: rewrite the last sentences focusing on Rwanda. It must be specific and precise, because your study is conducted in Rwanda. Show how severe the problem is in Rwanda.

3. Keywords: alcohol consumption; is it relevant?

4. Introduction: on page 4, line 89-94, minor revision is needed. For example: saying “Case-control study in Korea or Cross-sectional study in Tanzania” doesn’t interesting for reader. Not necessary to mention study type and design. Revise it in whole document.

5. Introduction [last paragraph]: the rationale or aim of research is not written clearly and briefly. It should be very simple and clear for reader. Plus that, it has to align with the research objective and title.

6. Study design: minor error: unmatched 1:2 case-control design =�unmatched case control [1:2]. Revise line 119 on page 5.

7. Exclusion criteria: women with a history of placental abnormalities were excluded. For instance: if the mother is get pregnant for the first time. She doesn’t have chance to be ruled out or to be examined. How did you handle such cases? Other, orthodontic braces: does it has association with pre-eclampsia? Why did you excluded them? Other, periodontal treatment prior 4 weeks; is 4 week is standard?

8. Ethical clearance: make it precise and clear. Major revision

9. Data collection process: this section needs major revision, Please make it precise and clear. This is not proposal, it’s a manuscript. Don’t put everything.

10. Data analysis: revise the first paragraph [Line 183-188, page 7] under data analysis section

11. Result section: this part needs major revision:

- The 1st sentences of the result part is not relevant. Please see other studies how begging the result section

- Subsection headings are mandatory [ for example: sociodemographic, associated factors]

12. Discussion: good. But it needs minor revision. Logical sequences of ideas. Plus that, making the sentences or paragraphs understandable for reader. See it again.

13. Confounders: how did you handled the confounders? It’s know that pre-eclampsia is multifactorial. You mentioned some of them on page-9, line 245-247. As we know like endothelial dysfunction, placental insufficient etc. has effect on pre-eclampsia development. How could you get to be sure or manage confounding effect in this cases? This needs very clear JUSTIFICATION!

14. Remove header the from all pages “Periodontitis and Pre-eclampsia in pregnant Women”. Not relevant.

6. PLOS authors have the option to publish the peer review history of their article (what does this mean?). If published, this will include your full peer review and any attached files.

Reviewer #1: **Yes: **Antonio Magan-Fernandez

Reviewer #2: No

---

## [Author Response · Author response to Decision Letter 0]

23 Apr 2024

Dear Editor, 

Thank you very much for the constructive comments from the reviewers. We have stringently addressed all comments in the file tiled "'Point-by-point response to reviewers' file 040424" as suggested by the Journal.

---

## [Decision Letter · Decision Letter 1]

16 Aug 2024

PONE-D-23-41485R1Periodontitis and Pre-eclampsia among Pregnant Women in Rwanda: A Case-Control Study.PLOS ONE

Dear Dr. Gatarayiha,

Thank you for submitting your manuscript to PLOS ONE. After careful consideration, we feel that it has merit but does not fully meet PLOS ONE’s publication criteria as it currently stands. Therefore, we invite you to submit a revised version of the manuscript that addresses the points raised during the review process.

**ACADEMIC EDITOR Comments: **

1. The introduction provides a broad overview of the literature but lacks a critical evaluation of the existing studies. Some of the references used are outdated or lack relevance to the current study's geographical context in the introduction and discussion sections.

2. The explanation of the sample size calculation using G*Power is incomplete.  The effect size of 0.50 is mentioned, but it is unclear what this value represents. Typically, in case-control studies, the effect size might refer to the odds ratio (OR) expected between exposure (e.g., periodontitis) and the outcome (e.g., pre-eclampsia). It would be helpful to include more details on the effect size and the assumptions made for the calculation.

3. There is a limited discussion with regards to other studies findings. Add other studies results in discussion section.

We look forward to receiving your revised manuscript.

Kind regards,

Feriha Fatima Khidri

Academic Editor

PLOS ONE

Reviewers' comments:

Reviewer's Responses to Questions

**Comments to the Author**

1. If the authors have adequately addressed your comments raised in a previous round of review and you feel that this manuscript is now acceptable for publication, you may indicate that here to bypass the “Comments to the Author” section, enter your conflict of interest statement in the “Confidential to Editor” section, and submit your "Accept" recommendation.

Reviewer #2: All comments have been addressed

Reviewer #3: (No Response)

Reviewer #4: All comments have been addressed

2. Is the manuscript technically sound, and do the data support the conclusions?

Reviewer #2: Yes

Reviewer #3: Partly

Reviewer #4: (No Response)

3. Has the statistical analysis been performed appropriately and rigorously? 

Reviewer #2: Yes

Reviewer #3: No

Reviewer #4: (No Response)

4. Have the authors made all data underlying the findings in their manuscript fully available?

Reviewer #2: Yes

Reviewer #3: Yes

Reviewer #4: (No Response)

5. Is the manuscript presented in an intelligible fashion and written in standard English?

Reviewer #2: Yes

Reviewer #3: Yes

Reviewer #4: (No Response)

6. Review Comments to the Author

Reviewer #2: Dear, Authors

It’s my great pleasure to be assigned as a reviewer of this research that focuses on “Periodontitis and Pre-eclampsia among Pregnant Women in Rwanda: A Case-Control Study,” which has great public health importance.

I found that you incorporated all the suggestions and comments I have given before. I am satisfied with all your suggestions. Still, there might be minor errors in a whole document. So you have to work on that to make your paper intelligible to the scientific community.

Reviewer #3: Periodontitis and Pre-eclampsia among Pregnant Women in Rwanda: A Case-Control

Study.

While the manuscript is very written as it is, here are a few concerns that needs to be addressed.

What is the specific prevalence of periodontitis in Rwanda, while the author provides reference for sub-Saharan African there is none specific to Rwanda.

Moreover Authors did not provide a convincing fine mechanistic relationship between Preeclampsia and Periodontitis in the manuscript. Thus, it is easy to presume that if oral hygiene is a public health concern in Rwanda, then finding a high prevalence of periodontitis in women with preeclampsia is also plausible

Perhaps as the author relate this two conditions to inflammation- additional inflammatory markers could have been analysed to provide novelty, a more solid association, and not only statistical inference.

Basic information on several factors used the model analysis have not been previously present in the manuscript. this should be presented and compared in both cohort before the logistic regression analysis.

Information on tobacco consumption in both cohort is not evident.

Information on oral hygiene is also not provided in the manuscript. these are important to contextualise the findings.

Although authors have indicated that Both conditions share many risk factors such as smoking, alcohol

consumption, diabetes, and maternal age among others, some of these risk factors were not assessed in the women. or were not documented in this manuscript.

The stated study limitations are within the remit of the study and authors should have performed these.

Reviewer #4: The authors have duly addressed all the comments raised in the thesis. This has tremendously improve the manuscript and put it in acceptable state. I have no further comment for the authors. I congratulate the for taking tine to address all the comments raised in the thesis

7. PLOS authors have the option to publish the peer review history of their article (what does this mean?). If published, this will include your full peer review and any attached files.

Reviewer #2: No

Reviewer #3: No

Reviewer #4: No

---

## [Author Response · Author response to Decision Letter 1]

13 Sep 2024

Dear Editor,

We wish to express our sincere appreciation to the editor and reviewers for their insightful comments. We have stringently addressed all comments in the file titled" point-by-point response to editor and reviewers file 12.09.24" as suggested by the journal.

We hope that the final version of this manuscript is much improved due to their comments.

---

## [Editor Report · Decision Letter 2]

1 Oct 2024

Periodontitis and Pre-eclampsia among Pregnant Women in Rwanda: A Case-Control Study.

PONE-D-23-41485R2

Dear Dr. Gatarayiha,

We’re pleased to inform you that your manuscript has been judged scientifically suitable for publication and will be formally accepted for publication once it meets all outstanding technical requirements.

Kind regards,

Feriha Fatima Khidri

Academic Editor

PLOS ONE
---

## [Editor Report · Acceptance letter]

4 Oct 2024

PONE-D-23-41485R2 

PLOS ONE

Dear Dr. Gatarayiha, 

I'm pleased to inform you that your manuscript has been deemed suitable for publication in PLOS ONE. Congratulations! Your manuscript is now being handed over to our production team.

Kind regards, 

on behalf of

Dr. Feriha Fatima Khidri 

Academic Editor

PLOS ONE